# The Revival of Aztreonam in Combination with Avibactam against Metallo-β-Lactamase-Producing Gram-Negatives: A Systematic Review of In Vitro Studies and Clinical Cases

**DOI:** 10.3390/antibiotics10081012

**Published:** 2021-08-20

**Authors:** Carola Mauri, Alberto Enrico Maraolo, Stefano Di Bella, Francesco Luzzaro, Luigi Principe

**Affiliations:** 1Clinical Microbiology and Virology Unit, “A. Manzoni” Hospital, 23900 Lecco, Italy; c.mauri@asst-lecco.it (C.M.); f.luzzaro@asst-lecco.it (F.L.); 2First Division of Infectious Diseases, Cotugno Hospital, AORN dei Colli, 80131 Naples, Italy; albertomaraolo@mail.com; 3Clinical Department of Medical, Surgical and Health Science, Trieste University, 34128 Trieste, Italy; stefano932@gmail.com; 4Clinical Pathology and Microbiology Unit, “S. Giovanni di Dio” Hospital, 88900 Crotone, Italy

**Keywords:** aztreonam, avibactam, ceftazidime/avibactam, *Enterobacterales*, *Pseudomonas*, *Stenotrophomonas*, metallo-β-lactamase, old antibiotic, antibiotic combination, last resource antibiotic

## Abstract

Infections caused by metallo-β-lactamase (MBL)-producing *Enterobacterales* and *Pseudomonas* are increasingly reported worldwide and are usually associated with high mortality rates (>30%). Neither standard therapy nor consensus for the management of these infections exist. Aztreonam, an old β-lactam antibiotic, is not hydrolyzed by MBLs. However, since many MBL-producing strains co-produce enzymes that could hydrolyze aztreonam (e.g., AmpC, ESBL), a robust β-lactamase inhibitor such as avibactam could be given as a partner drug. We performed a systematic review including 35 in vitro and 18 in vivo studies on the combination aztreonam + avibactam for infections sustained by MBL-producing Gram-negatives. In vitro data on 2209 Gram-negatives were available, showing the high antimicrobial activity of aztreonam (MIC ≤ 4 mg/L when combined with avibactam) in 80% of MBL-producing *Enterobacterales*, 85% of *Stenotrophomonas* and 6% of MBL-producing *Pseudomonas*. Clinical data were available for 94 patients: 83% of them had bloodstream infections. Clinical resolution within 30 days was reported in 80% of infected patients. Analyzing only patients with bloodstream infections (64 patients), death occurred in 19% of patients treated with aztreonam + ceftazidime/avibactam. The combination aztreonam + avibactam appears to be a promising option against MBL-producing bacteria (especially *Enterobacterales*, much less for *Pseudomonas*) while waiting for new antimicrobials.

## 1. Introduction

The global spread of metallo-β-lactamase-producing Gram-negatives (MBL-GN) is a serious cause of concern for public health. In particular, class B1 β-lactamases including Verona integron-encoded MBLs (VIM), imipenemases (IMP), and New Delhi MBLs (NDM), mostly carried by *Enterobacterales* and *Pseudomonas aeruginosa*, have now spread worldwide with multitudes of clinical variants [1]. The increase in population exchange at the global level and the prevalent plasmid-mediated nature, as well as the intestinal carriage of MBL-GN, contributed to the uncontrolled MBL-related resistance spread worldwide [2]. Invasive infections by MBL-GN are associated with high mortality rates (>30%), especially in the hospital setting when critically ill patients are involved [3,4]. MBL producers are mostly resistant to all β-lactams, including carbapenems and β-lactam/β-lactamase inhibitor combinations (BLBLICs). The optimal treatment of infections sustained by MBL-producing *Enterobacterales* or *P. aeruginosa* is not well defined, being associated with limited clinical experience and with few therapeutic alternatives (i.e., colistin, fosfomycin, cefiderocol) [4,5,6,7,8]. Moreover, the most recently approved BLBLICs (i.e., ceftazidime/avibactam, meropenem/vaborbactam, ceftolozane/tazobactam and imipenem/relebactam) are not active against MBL producers. This point is crucial when considering empirical treatment, since the detection of the MBL enzymes could not be determined a priori, but only after some days in most cases, when microbiological culture results are available. In sharp contrast with the novel BLBLICs, another combination shows promise against MBL-producing pathogens, namely the combination of avibactam (AVI) with aztreonam (ATM), which restores the in vitro activity and in vivo efficacy of ATM against these germs thanks to the inhibition of the prevalent co-carriage of non-MBL β-lactamases (e.g., ESBLs, AmpC-type enzymes) [9] (Figure 1). Pending the approval of this further new combination, the clinical use of the association of ATM and a BLBLIC such as ceftazidime-avibactam (CZA) has gained a foothold, becoming an attractive alternative for treating infections sustained by MBL producers [10]. Moreover, the introduction of this combination has made it possible to treat severe MBL-GN infections with β-lactams, thus overcoming issues related to the use of polymyxins, both in terms of pharmacokinetics/pharmacodynamics (poor lung penetration) and toxicity (renal injury) [11]. 

ATM is an old antibiotic that was approved by the United States (US) Food and Drug Administration (FDA) and European regulatory authorities in 1986. It was largely used in the past for the treatment of urinary tract, lower respiratory tract, and intra-abdominal infections, as well as septicemia, endometritis, pelvic cellulitis, and skin and skin structure infections due to aerobic Gram-negative organisms [12]. During the last three decades, its clinical use was strongly limited by the spread of ESBL and AmpC-type determinants. MBLs are able hydrolyze all β-lactams, except for the monobactam ATM. This specific characteristic currently defines the revival of this old antibiotic, highlighting the potential clinical use of ATM against MBL producers [13]. However, due to the frequent co-production of class A β-lactamases or AmpC-type determinants within MBL-GN, ATM remains active only in one-third of cases [1]. AVI is a novel non-β-lactam β-lactamase inhibitor with a potent activity against class A β-lactamases and AmpC-type determinants, available since 2015 in combination with ceftazidime (CAZ). Hence, clinicians used CZA to take advantage of the inclusion of AVI in the combination, thus protecting ATM from inhibition. A single product formulation of ATM-AVI is currently under development in Phase III studies for the treatment of serious infections (i.e., complicated intra-abdominal infections, nosocomial pneumonia including hospital-acquired pneumonia and ventilator-associated pneumonia, complicated urinary tract infections, or bloodstream infections) caused by MBL-producing Gram-negative bacteria. The combination of CZA plus ATM was also considered a preferred regimen for MBL-producing *Enterobacterales* by a recent IDSA guidance document [14]. However, clinical data on this combination are still scant, as well as information on resistance rate and resistance mechanisms among MBL-GN.

We performed a systematic review of the available observational literature on the clinical and microbiological features related to the use of ATM and AVI against MBL-GN, considering in vitro studies and clinical case reports. In particular, our analysis aimed to evaluate (and summarize): (1) all clinical case reports and in vitro studies in which ATM plus AVI was reported against MBL producers; (2) the characteristics of ATM plus AVI use (e.g., dosages and pharmacokinetic/pharmacodynamics targets); (3) the patient’s outcome; (4) the resistance rate; and (5) the involved resistance mechanism(s).

## 2. Results

### 2.1. Literature Search

The search strategy yielded 784 references; after de-duplication, 526 were excluded on the basis of title and abstract screening. Of the remaining 90 studies, 37 were excluded due to the reasons listed in Figure 2, wherein the entire process of article selection is illustrated. Overall, 35 in vitro studies [15,16,17,18,19,20,21,22,23,24,25,26,27,28,29,30,31,32,33,34,35,36,37,38,39,40,41,42,43,44,45,46,47,48,49] and 18 in vivo studies [4,47,50,51,52,53,54,55,56,57,58,59,60,61,62,63,64,65] were included.

### 2.2. General Features and Key Findings

#### 2.2.1. In Vitro Studies

Thirty-five articles regarding in vitro studies on antimicrobial activity of the combination ATM and AVI or CZA were evaluated, involving a total of 2209 MBL-producing isolates (Table 1), belonging to *Enterobacterales* (59.9%; n = 1324), *P. aeruginosa* (34.9%; n = 772), *Stenotrophomonas maltophilia* (3.4%; n = 76) and *Elizabethkingia anophelis* (1.7%; n = 37). When specified, bacterial species belonging to *Enterobacterales* were *Klebsiella pneumoniae* (n = 333), *Escherichia coli* (n = 59), *Enterobacter cloacae* (n = 9) and *Citrobacter freundii* (n = 1). ATM was tested in combination with AVI (alone) for 2044 isolates (92.5%) and in combination with CZA for 165 isolates (7.5%).

Bacterial isolates produced NDM-type enzymes (39.1%; n = 865; variants -1, -5, -6, -7, when specified), VIM-type (22.4%; n = 494; variants -1, -2, -4, -27, when specified) and IMP-type (7.6%; n = 168; variants -4, -8, -14, -48, when specified), taking into account that nine isolates co-harbored two MBLs. Moreover, basal MBLs were L1 produced by *S. maltophilia* isolates (3.4%; n = 76) and GOB and BlaB produced by *E. anophelis* (1.7%; n = 37). Not specified MBLs were reported for 579 isolates (26.2%), including 13 cases in which two or three not specified MBLs were produced. 

The high antimicrobial activity of ATM (MIC ≤ 4 mg/L) when combined with AVI or CZA was reported for 1167 (52.8%) isolates, almost entirely belonging to *Enterobacterales* (90.3%; n = 1054) and mostly producing NDM-type enzymes (60%; n = 700). 

Only cumulative MIC values of ATM in combination with AVI or CZA were reported for 527 (23.8%) isolates; hence, the specific antimicrobial activity of ATM cannot be evaluated against these strains. However, among these isolates, mostly synergistic interactions have been reported for *Enterobacterales*, while *P. aeruginosa* isolates mainly did not show synergistic interactions (Table 1). Overall, when reported, MIC values ≤4 mg/L for ATM in combination with AVI or CZA have been described in 79.6% of MBL-producing *Enterobacterales*, 85.5% of *S. maltophilia* (few strains, only one report) and 6.2% of MBL-producing *P. aeruginosa* isolates. 

The low antimicrobial activity of ATM (MIC values >4 mg/L, in combination) was reported for 808 (36.6%) isolates, almost totally belonging to *P. aeruginosa* (n = 724, 89.6%) (Table 2). For these isolates, ATM was tested in combination with AVI (alone) for 800 isolates (99%) and in combination with CZA for 8 isolates (1%). Notably, in a single study, 308 (38.1%) *P. aeruginosa* isolates have been reported as not synergistic to the combination ATM with AVI, but only cumulative MIC data were available (MIC50 and MIC90 were ≥16 and ≥32, respectively) [18]. Other isolates belonged to *Enterobacterales* (4.4%; n = 36), *E. anophelis* (4.6%; n = 37) and to *S. maltophilia* (1.4%; n = 11). Of note, isolates presenting low antimicrobial activity or not synergistic interactions produced VIM-type (34.4%; n = 278) enzymes (only one *E. coli* isolate), IMP-type (4.8%; n = 39) enzymes (*P. aeruginosa* only), NDM-type (4.3%; n = 35) enzymes (only three *P. aeruginosa* isolates), GOB and BlaB (4.6%; n = 37) enzymes (*E. anophelis* only) and L1 (1.4%; n = 11) enzymes (*S. maltophilia* only). For 408 (50.5%) isolates (n = 405 *P. aeruginosa*, n = 3 *Enterobacterales*), MBL types were not specified. The majority of MBL-producing *Enterobacterales* found to be less susceptible to ATM-AVI showed the co-production of other resistance determinants in various combinations, such as TEM-1, KPC-3, OXA-48, ESBLs (CTX-M-type, SHV-type), AmpC-type (FOX-type, CMY-type, DHA-type), OXA-1 and OXA-10, while *P. aeruginosa* isolates mainly showed impermeability (porin loss), production of PDC variants and OXA enzymes (other than OXA-48) or hyperexpression of efflux systems. Notably, 17 *E. coli* isolates had alterations (amino acid insertions) of the PBP3 structure (two of them also co-produced CTX-M-15, CMY-type and OXA-1 enzymes). 

When reported, MIC values for ATM alone ranged from 0.06 to 1024 mg/L. In *Enterobacterales*, MIC50 and MIC90 values of ATM in combination with AVI or CZA ranged from 0.125 to 0.25 and from 0.125 to 4 mg/L, respectively, hence highlighting, in most cases, a >128-fold reduction in ATM MIC values when tested in combination. Regarding *P. aeruginosa* isolates, MIC50 and MIC90 values were significantly higher than those reported in *Enterobacterales*, ranging from 16 to 32 and from 32 to 64 mg/L, respectively. In *S. maltophilia*, only 11/76 isolates showed MIC values of 8–64 mg/L. The highest MICs were reported by a single study in *E. anophelis*, with all isolates (n = 37) presenting MIC values >256 mg/L. 

Data were almost totally obtained by checkerboard assays (92.5% of tested isolates), using a fixed concentration of 4 mg/L for AVI (Table 1). 

#### 2.2.2. In Vivo Studies

A total of 18 studies were retrieved addressing the clinical use of ATM plus CZA against MBL-producers, whose general features are detailed in Table 3. The relative majority (7 out of 18) were conducted in the US [50,51,52,56,57,62,65]. The remaining were performed in the following countries: France (3), Italy (3), Australia (1), Greece (1), Mexico (1), Netherlands (1), Saudi Arabia (1), Spain (1) [4,47,53,54,55,58,59,60,61,63,64]. All of them were case reports/case series, barring two different cohorts (one retrospective, another prospective) of patients from Italy and Greece affected by NDM-producing isolates [4,47]. Of the latter patients, 64 received the combination under investigation.

Overall, of the 94 included patients, 97% (91/94) are adults [55,62,65]. Precise data on gender were available for 82 patients: 72% were male (59/82); in the retrospective Italian study, the overall cohort, comprising the 12 cases of interest, was made up of males mostly (70%, 28/40) [47]. The large majority of causative agents were represented by bacteria belonging to *Enterobacterales* (96%, 90/94, mostly *K. pneumoniae*, *E. coli* or *E. cloacae*). There were just three cases of non-fermenting Gram-negative bacteria involvement (*P. aeruginosa* and *S. maltophilia*) [50,51,62]. Bloodstream infections (BSIs), including the one related to central lines, were the most frequent type of clinical scenario (83%, 78/94). Various dosages of CZA and ATM were used, also modified according to renal function or pediatric age; the most represented was CZA 2.5 g plus ATM 2 g, both each eight hours (78%, 73/94). Duration of antibiotic therapy ranged from 10 to 49 days. 

The combination of CZA and ATM was administered as targeted therapy in nearly all cases (99%, 93/94), namely after demonstration of the synergistic activity of the association of the drugs, inactive when considered singularly. The only exception was a case of BSI by a presumptive MBL-producing strain: carbapenemase was not detected but suspicion was raised in light of medical history (previous treatments in India) and of the susceptibility profile of the *K. pneumoniae* isolate, but synergism was not demonstrated between CZA and ATM [56]. Outcomes’ definitions were quite heterogeneous. No adverse event related to CZA plus ATM treatment was reported. With regard to clinical efficacy, clinical resolution within 30 days was achieved in almost four-fifths of patients (80%, 75/94). Early recurrence was described in four cases [56,58,64]: notably, in three patients [58,64], the same treatment was re-administered, obtaining resolution of infection except in a single subject expired owing to chronic lung transplant rejection [58]. In the case series by Shah and colleagues, two late recurrences (within 90 days of follow-up) were described as well [56], and another late recurrence (at 4-month of follow-up) was described in the other case series [61]. Death by all causes occurred in 15 patients in early follow-up (within 30 days) and in another 2 patients when considering longer durations of monitoring after completion of antibiotic courses.

The two cohorts allowed a meta-analytic approach to compare, in the context of BSIs, the combination of CZA and ATM versus other available targeted therapies, based on the administration of at least one active agent, considering 30-day mortality as the endpoint (Figure 3). In total, 64 patients in each group were evaluable: death occurred in 19% of patients receiving CZA plus ATM (12/64) and in 44% of patients not receiving the aforementioned combination (28/64). Therefore, CZA plus ATM was associated with a lower 30-day mortality risk: both under fixed effect and random effect models, the OR was equal to 0.30 (95% CI, 0.13–0.66), and no heterogeneity was detected (I^2^ = 0%). Of note, in the comparator group, 30-day mortality crude rate was 58% (21/36) among patients being administered colistin-containing regimen and 25% (7/28) in subjects receiving regimens not including colistin (data not shown but abstracted from the primary studies).

### 2.3. Quality of Included Studies

#### 2.3.1. In Vitro Studies

The quality assessment of the in vitro studies is reported in Appendix A. The large majority of articles seemed to fulfill most of the qualitative criteria set by the tool adapted for in vitro studies (https://jbi.global/critical-appraisal-tools, last accessed 22 June 2021). The included studies seem to be a reliable source to describe the antimicrobial activity of the association between ATM and AVI or CZA against MBL-GN. Some studies do not provide a detailed description of isolation background and/or resistance determinants other than MBLs, since these aspects do not represent the central aim of the studies.

#### 2.3.2. In Vivo Studies

The quality assessment of the clinical studies (case report, case series, cohort study) is reported in Appendix A. From a qualitative perspective, the large majority of articles appeared to meet most of the criteria established by the different tools, adopted for each type of study design (https://jbi.global/critical-appraisal-tools, last accessed 22 June 2021). Therefore, the included studies seem to be a trustworthy source to describe the real-life use of the association between CZA and ATM to address difficult-to-treat infection by MBL-producing strains.

## 3. Discussion

The treatment of MBL infection has been a difficult challenge for at least two decades [66]. MBLs are capable of hydrolyzing all β-lactams, with the exception of ATM. At any rate, due to the frequent co-production of other β-lactamases within MBL-producing *Enterobacterales*, ATM is active against just about 30% of these isolates [23]. The therapeutic choice usually relied on individual susceptibility profile, resorting to agents such as aminoglycosides, tetracyclines, fosfomycin, and polymyxins, without an established therapeutic consensus and facing many safety issues [1]. New commercially available BLBLICs are inactive against MBL producers, while cefiderocol and the repurposing of old agents (e.g., intravenous fosfomycin) have expanded the armamentarium of potentially effective drugs against highly resistant Gram-negative bacteria [67], but the therapeutic options remain limited. A tentative algorithm for the empirical treatment of severe infections likely due to MBL-producing strains has been suggested by Bassetti and colleagues: a pre-eminent role, in light of their good safety profile coherent with the one of the β-lactam class, is given to cefiderocol and to the association of CZA and ATM [68]. The latter may exploit the activity of ATM against MBL and the activity of CZA against other β-lactamases, often co-existing and able to hydrolyze ATM. The combination of CZA and ATM derives from the current unavailability in clinical practice of AVI and ATM in a single product formulation, currently in Phase III. 

To sum up, this is the first review systematically describing the antimicrobial activity and the clinical use of ATM in combination with AVI or CZA against MBL-GN. In vitro studies have been conducted worldwide, with more than 2000 MBL-producing isolates tested. Overall, ATM in combination with AVI or CZA showed high antimicrobial activity against about 80% of MBL-producing *Enterobacterales*, mostly NDM producers, providing a >128-fold reduction in the MICs of ATM alone. The combination also showed high antimicrobial activity against isolates that co-produced other acquired resistant determinants, such as KPC and ESBLs. In line with these data, previously reported kinetic assays using purified protein extracts have demonstrated that AVI exerts potent activity against KPC, OXA-48, CTX-M-like and *E. cloacae* AmpC [69,70], protecting ATM by enzymatic hydrolysis related to these determinants. The combination was also highly active against 85% of *S. maltophilia* isolates and 6% of *P. aeruginosa* isolates. It is of note that almost all of *P. aeruginosa* isolates (>90%) showed MIC values ≥16 mg/L. These data provide impactful information to support clinical decisions about the use of ATM in combination with CZA when facing infections sustained by MBL-producing *Enterobacterales* and *S. maltophilia*, but not for MBL-producing *P. aeruginosa*. Notably, no MBL variants have been directly associated with increased MIC values for ATM in combination with AVI or CZA, neither in *Enterobacterales* nor *P. aeruginosa*, but low antimicrobial activity seems to be more likely related to other parameters. Overall, hyperexpression of efflux systems and/or the presence of derepressed *bla*_PDC_ variants, as well as the mutation in *oprD*, likely impact resistance to β-lactams in *P. aeruginosa*. In our search, in the case of *P. aeruginosa* isolates, an important role could be played by impermeability (porin loss), production of PDC variants, OXA enzymes (other than OXA-48) or hyperexpression of efflux systems. In particular, it is well-known that AVI may show a potent inhibitory activity against the basal AmpC produced by *P. aeruginosa*, even though extended-spectrum OXA enzymes can escape its wide spectrum of activity [71]. Selection of extended-spectrum OXA-2 or OXA-10 variants such as OXA-539, OXA-681 and OXA-14 has previously been associated with in vivo acquisition of high-level CZA resistance [72,73,74], while OXA-10 and OXA-18 enzymes have been associated with ATM’s high MIC values [75]. Moreover, previously reported development of resistance to CZA during treatment of *P. aeruginosa* infections has mainly been associated with selection of variants of PDC-enzymes [38,73]. Notably, co-production of PDC and OXA determinants conferring resistance to ATM in *P. aeruginosa* isolates has also previously been reported [76]. 

ATM in combination with AVI or CZA counts for a low antimicrobial activity or not synergistic interactions against only 3% of *Enterobacterales*. About 50% of them presented specific amino acid insertion (12 bp duplications) in PBP3 determinants after residue 333 (YRIN or YRIK), providing the reduction in molecular affinity for ATM [17,40]. Structural analysis suggests that this insertion will impact the accessibility of ATM (and other β-lactam drugs) to the transpeptidase pocket of PBP3 [17]. This particular polymorphism of PBP3 was associated with high MIC values for ATM in combination with AVI only in *E. coli* isolates. For the other remaining isolates, co-production and/or hyperexpression of ESBL and AmpC-type determinants could have contributed to the high MIC values of ATM in combination with AVI. Notably, the association of CMY-42 and the YRIK insertion in PBP3 has been demonstrated to confer resistance to ATM-AVI in *E. coli* by mutagenesis experiments [77]. ATM-AVI showed low antimicrobial activity against 15% of *S. maltophilia* isolates, even if potential resistance mechanisms were not investigated. Importantly, very low antimicrobial activity has also been reported against *E. anophelis*, with all isolates showing MIC values >256 mg/L for ATM-AVI. Since only one report exists, very few data are known about the efficacy of ATM-AVI against this species, as well as the possible contribution of GOB and BlaB to the resistance profile. This species, although sporadically reported, represents a very difficult-to-treat opportunistic pathogen, being resistant to all new commercially available BLBLICs (in addition to ATM-AVI) and allowing very limited therapeutic options for the treatment of related infections.

The use of ATM in combination with CZA is currently considered a reasonable option for clinical use in the management of infections sustained by MBL producers. The paramount issue is that the optimum pharmacokinetic/pharmacodynamic (Pk/Pd) target for ATM in combination with CZA is still to be determined. According to in vitro models, optimal eradication and resistance suppression may be achieved by administering 8 g instead of 6 g per day of ATM (as continuous or 2-h infusion each 6 h of 2 g) plus 2.5 g each 8 h of CZA [78]. Monte Carlo simulations run from a PK analysis of clinical samples of MBL-producing isolates from highly comorbid patients demonstrate that standard dosage (CZA 2.5 g and ATM 2 g every 8 h, the most frequent according to our systematic review) fulfilled the time-dependent Pd targets for these agents [79]. However, pediatric patients seemed to well tolerate very high dosages: for instance, 150 mg/kg/day for CZA (calculated on the basis of ceftazidime component) [55,65] and 100 mg/kg/day [55] or 150 mg/Kg/day for ATM [65], which would translate to more than 10 g daily for each drug in normal weight adult subjects. Notably, for the phase 3 study of ATM plus AVI, Pk data supported the selection of a maintenance dose equal to 6 g of ATM and equal to 2 g of AVI, both divided into four administrations every 6 h, after a loading dose equal to 500/167 mg (ATM and AVI, respectively), in patients with creatinine clearance >50 mL/min, for the Phase III development program [10].

Beyond the not negligible Pk/Pd issues, the question is whether current data permit the suggestion of CZA plus ATM against MBL infections preferentially over other available options. In the prospective international cohort described by Falcone and colleagues, a propensity score-based matched analysis, allowing a minimal loss of initial information (only two not-matched patients from the inception cohort of 102 subjects), showed that CZA plus ATM was associated with a lower 30-day mortality rate compared with other active agents for the treatment of MBL-producing *Enterobacterales* causing BSI: hazard ratio 0.31 (95% CI 0.15–0.66) [47]. This value overlaps with the OR for mortality emerging from the meta-analytic results in the present study, favoring the combination of CZA and ATM over alternative targeted therapies. Of note, in the described casuistry of nearly 100 patients affected by MBL infection in a wide array of clinical scenarios, this combination demonstrated a very good safety profile (neither adverse events nor treatment discontinuations were registered) along with a high success rate as targeted treatment, both as first-line and salvage option.

The strength of the present work is rooted in the strict inclusion criteria, extensive literature search, granular description of clinical use of CZA plus ATM on a case-by-case basis and the large amounts of data regarding in vitro studies. However, this study presents some limitations. From a clinical standpoint, it was based only on observational studies, preeminently case series or case reports. Large prospective studies are urgently needed to better understand the in vivo efficacy of the ATM-AVI combination. Inevitably, the present study inherits their intrinsic limitations, specifically selection bias, impossibility to appropriately account for potential lurking variables, huge heterogeneity and subjectivity of outcomes’ definitions as well as vast variability pertaining to CZA plus ATM use (indication, dose, duration, first-line or salvage treatment). The meta-analytic results comparing CZA plus ATM versus other targeted therapy against MBL-producing isolates (only *Enterobacterales*) responsible for BSI draw solely on two non-randomized studies, having small sample sizes and not originally conceived to weigh up different treatment options; therefore, no rock-solid evidence can be inferred on the superiority of one antibiotic regimen over another. Moreover, some in vitro studies provided cumulative data only, making it difficult (or impossible) to extrapolate data referring to specific MBL-producing isolates.

Notwithstanding, this study provides ‘real-world data’ that, if properly interpreted in the wider framework of the healthcare evidence ecosystem, may contribute to recommendations and guidelines when a higher source of information in the hierarchy of evidence is lacking. At any rate, further studies are needed to better define the clinical efficacy of CZA plus ATM, and only randomized clinical trials will provide high-quality evidence on the best therapeutic option for infections by MBL-producing strains.

## 4. Methods 

### 4.1. Protocol and Registration

This systematic review was conducted in accordance with the PRISMA (Preferred Reporting Items for Systematic Reviews and Meta Analyses) guidelines [80]. The review protocol was registered at the Prospero international prospective register of systematic reviews (registration no. CRD42020220888).

### 4.2. Literature Search Strategy

By using the PubMed and the Embase databases, searches for relevant articles were performed with the following items: “(aztreonam) AND (avibactam)”. Searches were limited to peer-reviewed articles published in English up to 31 October 2020. The search was updated to include further articles published until 31 May 2021. Moreover, the reference lists of reports identified by this search strategy were also hand-searched to select further relevant articles.

### 4.3. Study Selection

Two reviewers (LP and CM) independently screened the titles and abstracts to determine eligibility for full-text review. Inclusion criteria were the following: (i) studies published in full; (ii) as mentioned above, studies published in English language; (iii) in vivo and in vitro studies investigating the association between ATM and AVI (with or without other agents) against MBL producers under a clinical and microbiological standpoint, respectively. Studies were excluded if they were commentaries, editorials or review papers.

### 4.4. Data Extraction

After the initial screening, all potential eligible articles were independently reviewed using the full text by the same researchers for final inclusion. Two reviewers (CM, AEM) extracted relevant information from each included study by resorting to a standardized data extraction sheet and then proceeding to cross-check the results. The extracted data included: publication year, study period, geographical setting, relevant clinical and microbiological information. In detail, the former were: study design, sample size, age and gender of study participant(s), type(s) of infection, causative agent(s), resistance mechanism(s), antimicrobial susceptibility testing (AST) profile, therapeutic regimen(s), efficacy and safety outcomes as reported by each study. The latter were, beyond general features of the included studies: number of isolates, MBL determinants, minimum inhibitory concentration (MIC) range, antimicrobial agents tested (ATM plus CZA or AVI), methods to evaluate interactions, other resistant determinants. Since no susceptibility breakpoint for ATM-AVI exists, a current EUCAST Pk/Pd non-species-related susceptibility breakpoint for ATM (4 mg/L) has been arbitrarily taken as reference to assess low (≤4 mg/L) and high (>4 mg/L) antimicrobial activity (https://www.eucast.org/fileadmin/src/media/PDFs/EUCAST_files/Breakpoint_tables/v_11.0_Breakpoint_Tables.pdf; last accessed on 4 July 2021).

### 4.5. Strategy for Data Synthesis

A qualitative assessment and synthesis of the main characteristics of included studies was carried out. Key findings were tabulated. Meta-analyses were performed on comparable outcomes measured by at least two studies. For dichotomous data, a pooled odds ratio (OR) as a summary estimate was calculated with its 95% confidence interval (CI). A 2-sided *p*-value less than 0.05 and a 95% CI that did not cross 1 (OR) were considered statistically significant. Meta-analyses were carried out by using both the fixed effect and random effects DerSimonian and Laird methods, compared graphically with forest plots. Heterogeneity between studies was gauged by resorting to I² statistics. Statistical analyses were conducted by using the statistical software R, version 1.3.1093 (RStudio Team) and package ‘metafor’.

### 4.6. Quality Assessment

Included studies were critically appraised through the tools of the Joanna Briggs Institute, according to the study design (https://jbi.global/critical-appraisal-tools, last accessed on 22 June 2021). The tool named Checklist for Analytical Cross Sectional Studies was adapted for the quality assessment of in vitro studies. Definitions of case report(s), case series and cohort studies were predicated on the work of Dekkers and colleagues [81]. Since summary quality scores may yield misleading results, a global judgement on the methodological quality of included studies was considered more appropriate [82,83].

### 4.7. Ethics

Ethics committee approval was not a prerequisite in this case because the project used anonymized data as pertaining to clinical information and original studies that had already received proper institutional review board approval.

## 5. Conclusions

In this review, we resumed at large the in vivo and in vitro activity of the combination ATM-AVI against MBL-GN. Taken together, these data suggest that ATM in combination with AVI or CZA is an important therapeutic option against MBL-producing *Enterobacterales*, with very few isolates showing high MIC values and mostly providing a favorable outcome in treated patients. Importantly, the presence of ceftazidime has been demonstrated to not affect the in vitro antimicrobial activity of the combination ATM-AVI in *Enterobacterales* [45]. However, MBL-producing *P. aeruginosa* remains an important unsolved issue, and treatment alternatives for related infections mainly rely on colistin, (±fosfomycin) or cefiderocol. Accordingly, the use of the ATM-CZA combination in the treatment of infections sustained by MBL-producing *P. aeruginosa* remains to be elucidated, since very few data are available. This point represents a serious cause of concern, and new antimicrobial options against MBL-producing *P. aeruginosa* are urgently needed.

Waiting for the approval of the fixed combination between AVI and ATM as a single product formulation, the intriguing synergy involving the two drugs may be exploited in the context of clinical use of CZA and ATM association as a valid therapeutic strategy. Nevertheless, the optimal dosing strategy remains to be elucidated, and another challenge to take into account is the current unavailability of automated susceptibility testing for this combination.

## Figures and Tables

**Figure 1 antibiotics-10-01012-f001:**
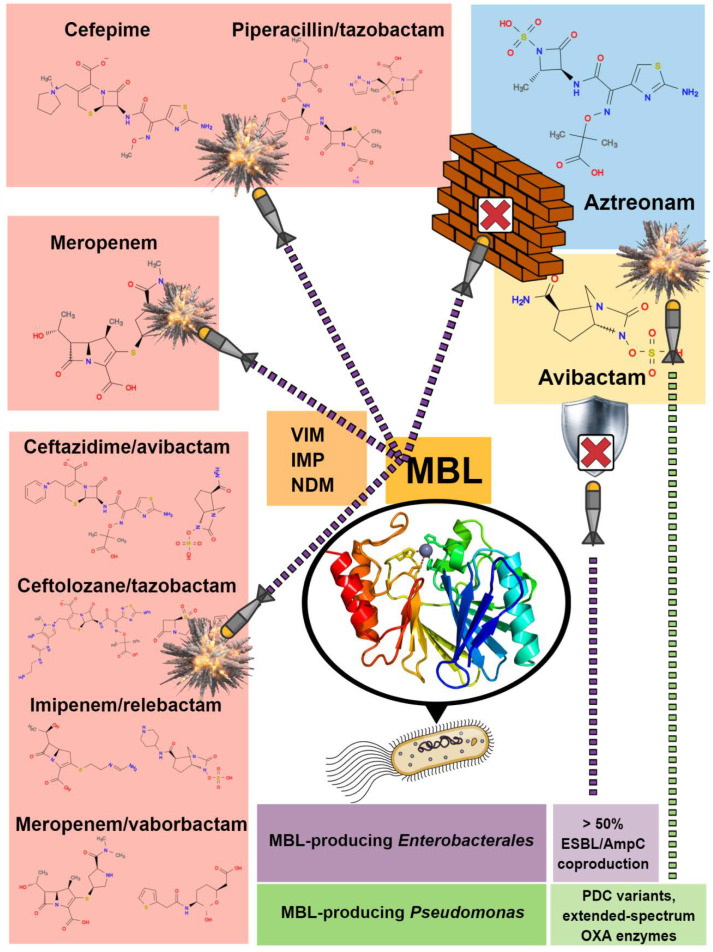
β-lactam antibiotic targets of MBL enzymatic activity.

**Figure 2 antibiotics-10-01012-f002:**
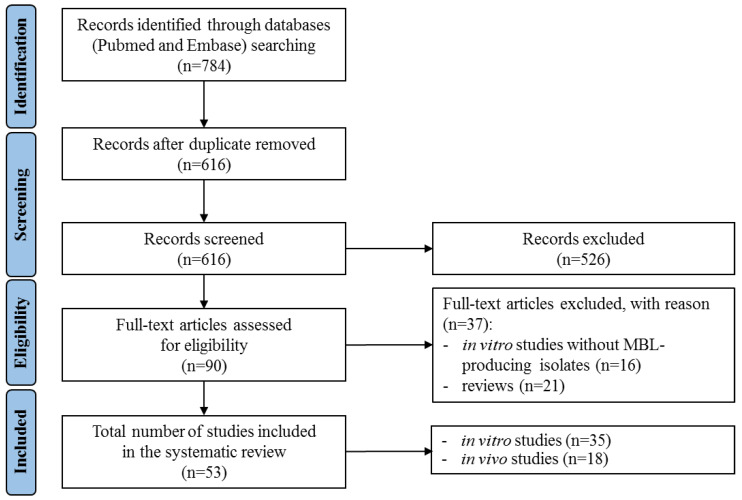
Literature search strategy.

**Figure 3 antibiotics-10-01012-f003:**
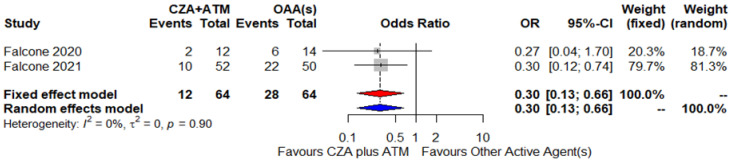
Forest plot of the 30-day mortality odds ratio between CZA plus ATM and other regimens against infections by MBL-producing strains. ATM, aztreonam; CZA, ceftazidime/avibactam; OAA, other active agent.

**Table 1 antibiotics-10-01012-t001:** In vitro studies on antimicrobial activity of ATM in combination with AVI or CZA against MBL-producing isolates.

Author and Publication Year (ref)	Region/Country or Type of Collection, Collection Period	MBL Isolates (n.)	MBL Determinants	ATM MIC (or MIC Range) (mg/L)	No. (%) of Isolates with MIC ≤4 mg/L for ATM in Combination with AVI	MIC range (mg/L) MIC_50_ (mg/L) MIC_90_ (mg/L) for ATM in Combination with AVI	Methods Used to Evaluate Interactions	Other Resistance Determinants	Antibiotic Combination	Notes
Livermore, 2011 [15]	NA	*Enterobacterales* (35)	NDM-1 (17)IMP-type (13)VIM-type (5)	0.06- >256 0.12- >256 0.25- >256	35 (100%)	All strains: ≤0.03–4; MIC_50_: 0.25; MIC_90_: 2; NDM: ≤0.03–4; IMP: 0.06–4; VIM: 0.12–0.5	CB	NA	ATM+AVI	-
Wang, 2014 [16]	China 2011–2012	*Enterobacterales* (14)	IMP-4 (10) NDM-1 (3) IMP-8 (1)	≤0.064- >128 MIC_50_: 128 MIC_90_: >128 (4 IMP-4-producing isolates were susceptible)	14 (100%)	≤0.064–0.5 MIC_50_: 0.125 MIC_90_: 0.5	CB	Most isolates co-produced various resistance determinants	ATM+AVI	-
Alm, 2015 [17]	USA, Philippines and India; collection period NA	*E. coli* (15) *K. pneumoniae* (12) *E. cloacae* (4)	NDM-type (15) NDM-type (12) NDM-type (4)	8- >512 32–512 256–512	13/15 (86.7%) 12/12 (100%) 4/4 (100%)	0.125–16 0.03–0.25 0.25–1	CB	All isolates co-produced a CMY-type enzyme; 13/15 isolates also produced ESBL determinants (mostly CTX-M-15)	ATM+AVI	Fourteen *E. coli* isolates presented a PBP3 insertion (MIC values of ATM in combination ranged from 4 to 16 mg/L. The last one *E. coli* (without insertion in PBP3) presented an ATM MIC value of 0.125 mg/L
Kazmierczak, 2015 [18]	Multiple sites around the world (40 countries) 2012–2014	*Enterobacterales* (163) *P. aeruginosa* (308)	NDM-type (72) VIM-type (64) IMP-type (27) VIM-type (270) IMP-type (35) NDM-type (3)	NDM ≤0.015- >128 MIC_50_: 128 MIC_90_: >128 VIM ≤0.015- >128 MIC_50_: 8 MIC_90_: 128 IMP 0.06- >128 MIC_50_: 64 MIC_90_: >128 VIM 0.5- >128 MIC_50_: 16 MIC_90_: 64 IMP 0.5- >128 MIC_50_: 16 MIC_90_: 64 NDM >128	NA (cumulative data only)	NDM ≤0.015- 8 MIC_50_: 0.12 MIC_90_: 0.5 VIM ≤0.015–2 MIC_50_: 0.12 MIC_90_: 1IMP 0.03–4 MIC_50_: 0.25 MIC_90_: 1 VIM 0.25- >128 MIC_50_: 16 MIC_90_: 32 IMP 2–128 MIC_50_: 32 MIC_90_: 64NDM 16- >128 (MIC range only)	CB	Most isolates also produced various resistance determinants	ATM+AVI	Mostly synergistic (cumulative data only, specific MIC values NA) Mostly not synergistic (cumulative data only, specific MIC values NA)
Li, 2015 [19]	Strains from not specified collection; collection period NA	*E. coli* (1) *E. cloacae* (1)	VIM-1 (2)	>64	0 (0%) 1 (100%)	8 0.12	CB	Both isolates co-produced TEM-1; *E. coli* co-produced KPC-3 variant	ATM+AVI	-
Vasoo, 2015 [20]	USA, Singapore and 1 strain from NCTC UK reference isolate NA	*Enterobacterales* (43) *P. aeruginosa* (4)	NDM-type (32) IMP-type (11) VIM-type (4)	0.12 to >512 ≤0.06 to >512 8	30 (93.8%) 11 (100%) 1 (25%)	NDM: ≤0.06 to 16; MIC_50_: 0.12;MIC_90_: 1; IMP: ≤0.06 to 1; MIC_50_: 0.25; MIC_90_: 1; VIM: 4 to 8; MIC_50_: 8; MIC_90_: 8	AD	NA	ATM+AVI	Two NDM-producing *E. coli* with ATM in combinations with AVI with MIC values at 8 (NDM-1) and 16 (NDM-7) mg/L
Pillar, 2016 [21]	NA	*E. coli* (1)	NDM-1	16	1 (100%)	0.12	CB	NA	ATM+AVI	-
Thomson, 2016 [22]	Kentucky (USA); collection period NA	*E. cloacae* (1)	VIM-1	>64	1 (100%)	0.5	CB	The isolate co-produced a KPC-18 variant	ATM+AVI	Synergistic interaction
Karlowsky, 2017 [23]	Worldwide (40 countries) 2012–2015	*Enterobacterales* (267) *P. aeruginosa* (452)	NDM-type (142) VIM-type (96) IMP-type (29) NA	≤0.015- >128 MIC_50_: 64 MIC_90_: >128 0.25- >128 MIC_50_: 16 MIC_90_: 64	265 (99.3%) 47 (10.4%)	≤0.015–8 MIC_50_: 0.12 MIC_90_: 1 0.25- >128 MIC_50_: 16 MIC_90_: 32	CB	Most isolates also produced various resistance determinants	ATM+AVI	-
Marshall, 2017 [24]	Not specified collection; Collection period NA	*Enterobacterales* (21)	NDM-1 (20) IMP-type (1)	0.0625->512 (17/21 isolates were resistant)	0 (0%)	8–64	AD with ATM fixed at 8, 16, 32 and 64	Most isolates also produced various resistance determinants	ATM+CZA	Mostly synergistic interactions (20/21) with ATM fixed at 8, 16, 32, and 64 mg/L
Wenzler, 2017 [25]	NA	*E. coli* (1) *P. aeruginosa* (1) *C. freundii* (1)	NDM-type (1) IMP-type (1) VIM-type (1)	>256 32 8	NA	NA	GDS	NA	ATM+CZA	Synergistic interactions for *E. coli* and *C. freundii* (FIC index ≤ 0.5), indifferent interaction for *P. aeruginosa* (0.5 < FIC index < 1)
Zhang, 2017 [26]	Indiana (USA) 2010–2013	*E. cloacae* (3) *K. pneumoniae* (1)	VIM-1 (3) NDM-1 (1)	NA	4 (100%)	0.12–0.25	CB	All isolates co-produced KPC-3 variant	ATM+AVI	-
Avery, 2018 [27]	AR Bank collection; collection period NA	*E. coli* (5) *K. pneumoniae* (5)	NDM-1 (2) NDM-5 (2) NDM-6 (1) NDM-1 (3) VIM-1 (1) IMP-4 (1)	64- >256	10 (100%)	NA	GDS	All isolates co-produce at least one serine-β-lactamase	ATM+CZA	*E. coli:* 4 isolates showed synergistic interactions (FIC index ≤ 0.5), 1 isolate showed additive interaction (0.5 < FIC index < 1)
Jayol, 2018 [28]	France, Colombia and Turkey; collection period NA	*K. pneumoniae* (15)	NDM-type (15)	0.19- >256 MIC_50_: >256 MIC_90_: >256	15 (100%)	0.094–2 MIC_50_: 0.38 MIC_90_: 1.5	GDS	Six isolates co-produced OXA-48 enzymes; 1 isolate co-produced OXA-181 enzyme; all strains were resistant to colistin	ATM+CZA	-
Sader, 2018 [29]	Worldwide, 2016	*Enterobacterales* (61)	NDM-1 (41)NDM-type (10) VIM-1 (7) IMP-type (3)	NA	61 (100%)	≤0.03 to 4 MIC_50_: 0.12MIC_90_: 0.5	CB	NA	ATM+AVI	-
Biagi, 2019 [30]	NA	*E. coli* (4) *K. pneumoniae* (4)	NDM-type (8)	0.25- ≥256 MIC_50_: 128 MIC_90_: ≥256	7 (87.5%)	≤0.03–16 MIC_50_: 0.25 MIC_90_: 16	CB, TK	All isolates co-produce at least one serine-β-lactamase	ATM+CZA	Time kill: synergistic for seven strains, except for the ATM-susceptible *E. coli*
Lin, 2019 [31]	China 2015–2016	*K. pneumoniae* (23)	NDM-1 (23)	MIC values NA; all strains were resistant	21 (91.3%)	≤0.25–8 MIC_50_: 0.5 MIC_90_: 2	CB	All isolates co-produced TEM-1 and SHV-12; most strains also have other resistance determinants	ATM+AVI	-
Mikhail, 2019 [32]	NA	*P. aeruginosa* (2)	IMP-48 (2)	64	0 (0%)	64	CB	OprD loss, OXA-10, MexCD-OprN overexpression (5X; only one isolate)	ATM+CZA	Not synergistic interactions
Pragasam, 2019 [33]	India, NA	*K. pneumoniae* (9)	NDM-type (9)	128- >1024	9 (100%)	≤0.12–0.25; MIC_50_: ≤0.12; MIC_90_: ≤0.12	CB	Six isolates co-harbored OXA-48-type enzymes	ATM+CZA	-
Zou, 2019 [34]	China 2012–2018	*Enterobacterales* (13)	NDM-5 (5) NDM-1 (4) IMP-4 (3) IMP-8 (1)	NA	12 (92.3%)	<=0.125–8 MIC_50_: 0.5 MIC_90_: 4	CB	All isolates co-produced ESBLs and presented porin (OMP) loss; the IMP-4-producing isolate co-produced a KPC-2 variant	ATM+AVI	-
Esposito, 2020 [35]	Africa/Middle east, Asia-Pacific, Europe and Latin America 2016–2017	*K. pneumoniae* (114)	NA	0.06- ≥256 MIC_50_: ≥256 MIC_90_: ≥256	114 (100%)	0.03–0.5 MIC_50_: 0.12 MIC_90_: 0.25	CB	NA	ATM+AVI	-
Kilic, 2020 [36]	Turkey; collection period NA	*K. pneumoniae* (17)	NDM-1 (13) NDM-1 + OXA-48 (4)	≥64	17 (100%)	<4; MIC_50_ and MIC_90_ values were NA	CB	NA	ATM+AVI	Synergistic for all isolates (FIC index ≤0.5)
Kim, 2020 [37]	Korea 201 4–2018	*K. pneumoniae* (11)	NDM-type	256- >512 MIC_50_: ≥512 MIC_90_: ≥512	11 (100%)	0.06–2 MIC_50_: 0.25 MIC_90_: 1	CB	NA	ATM+AVI	Two isolates showed MIC values of 64 mg/L for ATM in combination with AVI when tested at high bacterial inoculum (1 × 10^7^ CFU/mL)
Lee, 2020 [38]	FDA-CDC Bank collection, collection period NA	*P. aeruginosa* (5)	VIM-2 (3) VIM-4 (1) IMP-14 (1)	16- >128	0 (0%)	16–64 MIC_50_: 32 MIC_90_: 64	CB, TK	All isolates co-produce at least one serine-β-lactamase	ATM+CZA	Mostly synergistic interactions (4/5) with MIC values >4 mg/L for ATM in combination
Niu, 2020 [39]	HMH-CDI collection, NA	*K. pneumoniae* (68)	Single MBLs (55): NDM-type (43) VIM-type (11) IMP-type (1) Dual MBLs (12) Triple MBLs (1)	≤0.5 to >128 (65/68 strains were resistant)	66 (97.1%)	All strains: ≤0.25 to 8; MIC_50_: ≤0.25; MIC_90_: 1; Single MBL: ≤0.25 to 1; MIC_50_: ≤0.25; MIC_90_: 0.5; Dual/triple CPE: ≤0.25 to 8; MIC_50_: 0.5; MIC_90_: 8	CB	The two NDM-1-producing isolates with ATM MIC value of 8 mg/L produced NDM-1, OXA-48, CTX-M-15, CMY-16, SHV-1, TEM-1, OXA-10, SCO-1 associated with outer membrane protein defects (OmpK35, OmpK36)	ATM+AVI	Synergistic interactions
Periasamy, 2020 [40]	India 2016–2018	*E. coli* (23)	NDM-type (23)	>32	10 (43.5%)	2–16	CB	All isolates presented insertions in PBP3	ATM+AVI	-
Wei, 2020 [41]	Southwest China 2018–2019	*Enterobacterales* (26)	NDM-5 (14) NDM-1 (11) IMP-4 (1)	2–512	26 (100%)	0.125–2; MIC_50_: 0.125 MIC_90_: 2	CB	Four NDM-1-producing isolates co-harbored KPC genes	ATM+AVI	-
Yang, 2020 [42]	Taiwan 2012–2015	*K. pneumoniae* (14) *E. coli* (9)	*K. pneumoniae*: IMP-type (9) VIM-type (5) *E. coli*: NDM-type (5) VIM-type (3) IMP-type (1)	*K. pneumoniae*: 0.125- >32 (11/14 were resistant) *E. coli*: 2- >32 (8/9 were resistant)	*K. pneumoniae*: 13 (92.9%) *E. coli*: 9 (100%)	*K. pneumoniae*: <0.06- >32 MIC_50_: 0.125 MIC_90_: 0.5 *E. coli*: <0.06- 2 MIC_50_: 0.125 MIC_90_: 2	AD	All isolates also produced ESBL genes	ATM+AVI	-
Zhang, 2020 [43]	China October 2016–September 2017	*Enterobacterales* (161)	NDM-type (151) IMP-type (13) VIM-type (2)	≤1- ≥64 MIC_50_: ≥64 MIC_90_: ≥64	161 (100%)	≤0.125–4 MIC_50_: ≤0.125 MIC_90_: 1	CB	Most isolates also produced various resistance determinants	ATM+AVI	Five isolates co-harbored two MBLs
Zou, 2020 [44]	Southwest China, 2018–2019	*Enterobacterales* (29)	NDM-type (24) IMP-4 (1) NDM-1 (3) NDM-1+VIM-4 (1)	NA	29 (100%)	≤0.125 to 4 MIC_50_: ≤0.125 MIC_90_:1 (Cumulative data, not related only to MBL isolates)	CB	Three NDM-1-producing isolates co-harbored a KPC-2 gene	ATM+AVI	-
Bhatnagar, 2021 [45]	CDC & FDA Antibiotic Resistance Isolate Bank or CDC’s internal collection; collection period NA	*Enterobacterales* (37)	NDM-type (2) NDM-1 (24) NDM-5 (2) NDM-6 (1) NDM-7 (1) VIM-1 (2) VIM-27 (2) IMP-4 (2) IMP-8 (1)	8- >64	32 (80%)	≤0.5–8 MIC_50_: ≤0.5 MIC_90_: 8	CB	Five NDM-producing *K. pneumoniae* and one NDM-producing *E. coli* also co-produced OXA-232 and OXA-181, respectively	ATM+AVI	Five NDM-producing *E. coli* showed MIC values of 8 mg/L for ATM in combination with AVI
Chang, 2021 [46]	China 2015–2019	*Elizabethkingia anophelis* (37)	GOB and BlaB (basal enzymes)	>256	0 (0%)	>256	CB	35 isolates co-produced CME determinant	ATM+AVI	-
Falcone, 2021 [47]	Italy and Greece, November 2018–December 2019	*Enterobacterales* (52)	NDM-type (47) VIM-type (5)	NA	NA	NA	DD GDS	NA	ATM+CZA	Synergistic interactions
Lin, 2021 [48]	China 2011–2018	*Stenotrophomonas maltophilia* (76)	L1 (basal enzymes)	1–1024	65 (85.5%)	0.06–64 MIC_50_: 2 MIC_90_: 8	CB	All isolates co-produced L2 determinant (basal enzyme)	ATM+AVI	-
Maraki, 2021 [49]	Greece 2016–2020	*K. pneumoniae* (40)	NDM-type (35) VIM-type (2) NDM-type+VIMP-type (3)	24- >256	40 (100%)	0.06–0.56 MIC_50_: 0.31 MIC_90_: 0.37	GDS	KPC-type enzymes co-produced by one NDM-producing and two VIM-producing isolates	ATM+CZA	-

Abbreviations: AD, agar-dilution assay (with avibactam at a fixed concentration of 4 mg/L); ATM, aztreonam; AVI, avibactam; CZA, ceftazidime/avibactam; CB, checkerboard assay (with avibactam at a fixed concentration of 4 mg/L); DD, double-disk synergy test; GDS, gradient diffusion strip assay; TK, time-kill assay; NA, not available.

**Table 2 antibiotics-10-01012-t002:** Studies reporting low antimicrobial activity (MIC value > 4 mg/L) of ATM in combination with AVI or CZA against MBL-producing isolates.

Author and Publication Year (ref)	Number and Species (%) of Strains with MIC >4 mg/L for ATM+AVI	MBL Determinant	Presumptive Determinants Involved in High MIC Values for ATM in Combination	MICs of ATM in Combination with AVI and Supplemental Notes	Antibiotic Combination
Alm, 2015 [17]	2/31 (6.5%); *E. coli*	NDM-1	Both *E. coli* isolates presented a PBP3 (YRIK) insertion and co-production of other resistance determinants: 1) CTX-M-15, CMY-42, TEM-1, OXA-1; 2) CTX-M-15, CMY-4, OXA-1	14 *E. coli* isolates presented a PBP3 insertion but only two with MIC values >4 mg/L for ATM in combination with AVI	ATM+AVI
Kazmierczak, 2015 * [18]	308 (100%) *P. aeruginosa*	VIM-type (n = 270) IMP-type (n = 35) NDM-type (n = 3)	Co-production of various (not specified) resistance determinants (cumulative data)	Mostly not synergistic; MIC_50_ ≥16 mg/L MIC_90_ ≥32 mg/L (cumulative data)	ATM+AVI
Li, 2015 [19]	1/2 (50%) *E. coli*	VIM-1	Loss of an outer membrane protein; co-production of KPC-3 and TEM-1	MIC 8 mg/L	ATM+AVI
Vasoo, 2015 [20]	5/47 (10.6%) isolates: 2/32 (6.3%) *E. coli*; 3/4 (75%) *P. aeruginosa*	NDM-1 and NDM-7 (n = 2; *E. coli*) VIM-type (n = 3; *P. aeruginosa*)	Altered PBP3 affinity (*E. coli*); porin loss and overexpression of efflux pump (*P. aeruginosa*)	*E. coli* isolates with MIC values for ATM in combination with AVI MIC values of 8 (NDM-1) and 16 (NDM-7) mg/L, while *P. aeruginosa* isolates with MIC value of 8 mg/L	ATM+AVI
Karlowsky, 2017 [23]	2/267 (0.7%) *Enterobacterales* 405/452 (89.6%) *P. aeruginosa*	NA (cumulative data)	NA	*Enterobacterales:* MICs 8 mg/L *P. aeruginosa* (cumulative data)*:* MIC_50_ 16 mg/L MIC_90_ 32 mg/L	ATM+AVI
Marshall, 2017 [24]	4/21 (19%) isolates: *E. coli* (n = 2), *K. pneumoniae*, *P. rettgeri*	NDM-1 (n = 4)	Other co-produced determinants: *E. coli—*CTX-M-15, CMY-2, TEM-type; *K. pneumoniae—*CTX-M-15, CMY-2, DHA-type, SHV-type, TEM-type; *P. rettgeri—*CMY-2, DHA-type	All four isolates had MICs ≥ 16 mg/L. One *E. coli* showed synergy with ATM fixed at 32 mg/L, in combination with AVI; one *E. coli* and *K. pneumoniae* showed synergy with ATM fixed at 64 mg/L. *P. rettgeri* was not evaluated with ATM fixed at 32 and 64 mg/L	ATM+CZA
Wenzler, 2017 [25]	1/3 (33.3%) *P. aeruginosa*	IMP-type	NA	NA	ATM+CZA
Biagi, 2019 [30]	1/8 (12.5%) *Enterobacterales* (*E. coli*)	NDM-type,	Co-production of CMY-2/FOX-type, CTX-M-1, TEM-type	MIC 16 mg/L	ATM+CZA
Lin, 2019 [31]	2/23 (8.7%) *K. pneumoniae*	NDM-1	Co-production of TEM-1 and SHV-12	MICs 8 mg/L	ATM+AVI
Mikhail, 2019 [32]	2/2 (100%) *P. aeruginosa*	IMP-48	OprD loss, OXA-10, MexCD-OprN overexpression (5X; only one isolate)	MICs 64 mg/L	ATM+CZA
Zou, 2019 [34]	1/12 (8.3%) *Enterobacterales*	NDM-5	Loss of an outer membrane protein and co-production of ESBL determinant	MIC 8 mg/L	ATM+AVI
Lee, 2021 [38]	5/5 (100%) *P. aeruginosa*	IMP-14; VIM-4; VIM-2; VIM-2; VIM-2	Co-production of: OXA-10, OXA-488, VEB-9, PDC-2; OXA-396, PDC-3; OXA-488, PDC-2; OXA-488, PDC-3; PDC-8	MIC range 16–64 mg/L	ATM+AVI
Niu, 2020 [39]	2/68 (2.9%) isolates: *K. pneumoniae*	NDM-1	NDM-1, OXA-48, CTX-M-15, CMY-16, SHV-1, TEM-1, OXA-10, SCO-1 associated with outer membrane protein defects (OmpK35, OmpK36)	MICs 8 mg/L	ATM+AVI
Periasamy, 2020 [40]	13/23 (56.5%) *E. coli*	NDM-type	Insertions in PBP3 (YRIN and YRIK)	MICs 8–16 mg/L	ATM+AVI
Yang, 2020 [42]	1/14 (7.1%) *K. pneumoniae*	NA	NA	MIC 32 mg/L	ATM+AVI
Bhatnagar, 2021 [45]	5/37 (20%) *E. coli*	NDM-1 (3), NDM-5 (1), NDM-6 (1)	NA	MICs 8 mg/L	ATM+AVI
Chang, 2021 [46]	37/37 (100%) *Elizabethkingia anophelis*	GOB and BlaB (constitutionally produced enzymes)	GOB, BlaB and CME determinants	MICs >256 mg/L	ATM+AVI
Lin, 2021 [48]	11/76 (15.5%) *Stenotrophomonas maltophilia*	L1 (constitutionally produced enzyme)	NA	MIC range 8–64 mg/L	ATM+AVI

Abbreviations: ATM, aztreonam; AVI, avibactam; CZA, ceftazidime/avibactam; NA, not available. * The study of Kazmierczak et al. (2015) [18] presented only MIC cumulative data regarding 308 *P. aeruginosa* isolates; hence, MIC values of single isolates were not available. All isolates were considered to present not synergistic interactions. Taking into account the prevalent not synergistic interactions among tested isolates and their MIC_50_ and MIC_90_ values (>4 mg/L), these isolates have been aggregated to those reporting weak antimicrobial activity of ATM in combination with AVI.

**Table 3 antibiotics-10-01012-t003:** Overview of the studies’ features and report of the outcomes.

Authors and Publication Year (ref)	Study Design	Country and Time Span	Number of Patients (and Age in Years, Mean) ^a^	Sex (M, F, n)	Types of Infection (n)	Microorganism (n)	Mechanism of Resistance	AST Profile	Therapeutic Regimen (n)	Type and Line ^b^ of Therapy (n)	Outcome	Notes
Mojica, 2016 [50]	Case report	United States, NR	1 (19)	1F	CLABSI (1)	*Stenotrophomonas maltophilia* (1)	L1 and L2 β-lactamases	Resistance to CAZ, CZA, ATM, IMI	CZA 2.5 g tid + ATM 2 g tid for 48 days (1)	Targeted (1) – Salvage (1)	Clinical resolution without AE and no recurrence after ≥90 days of follow-up	Renal transplant patient. CZA + ATM started after 63 days of previous antibiotics
Davido, 2017 [51]	Cases report	United States, NR	2 (61)	1M, 1F	CLABSI (1), pneumonia with lung abscess (1)	*Klebsiella pneumoniae* (1), *Pseudomonas aeruginosa* (1)	CLABSI→OXA-48, NDM-1 Pneumonia→NDM-1, AmpC-hyperproducing	Both strains→Resistance to ATM, CZA (susceptibility only to COL and AMK)	CZA 2.5 g tid + ATM 2 g tid for 10 days (1, CLABSI); CZA 2.5 g bid + ATM 2 g bid for 42 days (1, pneumonia)	Targeted (2)–First line (1, pneumonia) Salvage (1, CLABSI)	CLABSI→resolution of infection, later death due to heart failure; Pneumonia→clinical resolution	In CLABSI patient, CZA + ATM started after 26 days of previous antibiotics
Mittal, 2018 [52]	Case report	United States, 2017	1 (42)	1M	BJI (1)	*Klebsiella pneumoniae* (1)	NDM-1 and OXA-181	Resistance to ATM, CZA, MER (susceptibility only to COL)	CZA 2.5 g tid + ATM 2 g tid for 48 days (1)	Targeted (1) – Salvage (1)	Microbiological cure and wound healing (functional limitation of the arm)	Right elbow osteomyelitis in a returning traveler from Bangladesh. Concurrent osteomyelitis from *Rhizopus*
Shaw, 2018 [53]	Case series	Spain, January 2016–June 2017	10 (68.5)	7M, 3F	BSI (4), HAP (2), cUTI (2), CLABSI (1), mediastinitis (1)	*Klebsiella pneumoniae* (10)	NDM-1 and OXA-48 as well as CTX-M-15	Resistance to all β-lactams, intermediate to TIG; in 40% of case resistance also to COL	CZA 2.5 g tid + ATM 3 g od (4); CZA 2.5 g tid + ATM 1 g tid (2); CZA 2.5 g tid + ATM 2 g tid (1); CZA 1.5 g tid + ATM 3 g od (1); CZA 1.25 g od + ATM 2 g od (1); CZA 940 mg bid + ATM 2 g od (1)	Targeted (10)–First line (5) Salvage (5)	At day 30, success rate 60%, 3 deaths and 1 recurrence (cUTI); among the 6 successful cases, 2 recurrences at day 90 (1 BSI, 1 cUTI). No AE and no attributable deaths related to infections.	Hospital outbreak started in late 2015 by an XDR strain (KP-HUB-ST147). Therapy duration: 3–28 days
Emeraud, 2019 [54]	Case report	France, NR	1 (70)	1M	cUTI (1)	*Escherichia* *coli* (1)	NDM-1 and OXA-48 as well as CTX-M-15	Extremely drug-resistant strain	CZA 2.5 g tid + ATM 2 g tid for 10 days (1)	Targeted (1) – Salvage (1)	Clinical resolution without AE and no recurrence over 2 months	Medical history of recent travel in India
Hobson, 2019 [55]	Case report	France, NR	1 (3)	1F	BSI (1)	*Morganella morganii* (1)	NDM-1	Resistance to CAZ, CZA, ATM, IMI	CZA 100 mg/Kg/day (as for ceftazidime component) + ATM 150 mg/Kg/die for 10 days (1)	Targeted (1) – First line (1)	Clinical resolution without AE and no recurrence over 6 months days of follow-up	Pediatric patient with lymphoblastic acute leukemia
Shah, 2019 [56]	Case report	United States, NR	1 (80)	1M	BSI (1)	*Klebsiella pneumoniae* (1)	Suspected but not confirmed carbapenemase (molecular testing not performed)	Susceptibility only to COL and TIG	CZA 0.94 g bid + ATM 1 g tid for 10 days (1)	Empiric (1)– Salvage (1)	Only partial improvement	BSI from urinary source. Therapy included POLB as third agent. Previous exposure to COL in India
Stewart, 2019 [57]	Case report	United States, NR	1 (61)	1M	Necrotizing kidney allograft infection (1)	*Enterobacter cloacae* (1)	NDM-1	Susceptibility only to COL	CZA + ATM for 46 days (1)	Targeted (1) – Salvage (1)	Infection resolution but patient dialysis- dependent at 4-month follow-up	Surgical source control was carried out. Concurrent fungal infection of the kidney with *Rhizopus*
Benchetrit, 2020 [58]	Cases report	France, NR	2 (55.5)	2F	BSI (1), VAP (1)	*Klebsiella pneumoniae* (2)	NDM-1	Susceptibility only to COL	CZA 0.94 g od + ATM 2 g bid for 14 days (1, BSI); CZA 2.5 g od + ATM 2 g bid for 14 days (1, VAP)	Targeted (2) – Salvage (2)	Resolution of infection, but in both cases recurrence within 30 days, treated with CZA+ATM; the patient affected by BSI survived, the other one died due to chronic lung transplant rejection	The cases described involved 2 solid organ transplant recipients (kidney and lung). In the case of VAP, TIG was added
Falcone, 2020 [4]	Retrospective cohort	Italy, November 2018–May 2019	NAAG	NAAG	BSI (12)	*Klebsiella pneumoniae* (NAAG), *Escherichia coli* (NAAG)	NDM-1 and NDM-5	NAAG	CZA 2.5 g tid + ATM 2 g tid	Targeted (12) – NAAG	Two patients died out of 12	Original cohort made up of 40 patients, 70% males, median age 70.5 (IQR 55.3–77.8), susceptibility rate of 91.4% to COL (out of 35 available strains). In the whole cohort, Charlson comorbidity index score and age were factors independently associated with 30-day mortality at Cox regression analysis
Alghoribi, 2021 [59]	Case report	Saudi Arabia, February 2019	1 (40)	1F	CIED/IE (1)	*Klebsiella pneumoniae* (1)	NDM-1, OXA-48, CTX-M-14b, SHV-28 and OXA-1	Pan-resistance	CZA 2.5 g tid + ATM 2 g tid for 46 days (1)	Targeted (1)– Salvage (1)	Infection resolution and no recurrence at 6-month follow-up	IE affecting the tricuspid valve along with CIED. Source control obtained through pacer device and lead tip replacement
Bocanegra-Ibarias, 2021 [60]	Case report	Mexico, NR	1 (35)	1M	BSI (1)	*Klebsiella pneumoniae* (1)	NDM-1	Susceptibility only to COL	CZA 2.5 g tid + ATM 2 g tid for 10 days (1)	Targeted (1)– Salvage (1)	Infection resolution and no signs or symptoms of infection after 80 days of follow-up, but subsequent death related to underlying disease	The patient underwent HSCT to treat a severe form of aplastic anemia
Cairns, 2021 [61]	Case series	Australia, NR	4 (59)	4M	BJI (1), CLABSI (1), cUTI (2)	*Enterobacter cloacae (4)*	IMP-4	Resistance to all β-lactams	CZA 2.5 g tid + ATM 2 g tid (3); CZA 0.94 g bid +ATM 1.5 g tid (1)	Targeted (4)– Salvage (4)	Infection resolution in all cases; one recurrence of cUTI at 4-month follow-up and one death not related to CLABSI beyond 4 weeks of follow-up	Outbreak in a single institution involving three SOT patients and one HSCT subject. Therapy duration: 24–49 days.
Cowart, 2021 [62]	Case report	United States	1 (11)	1F	Pneumonia (1)	*Stenotrophomonas maltophilia (1)*	L1 and L2 β-lactamases	Susceptibility only to MIN and TMP/SMX	CZA 150–200 mg/Kg tid (as for ceftazidime component) + ATM 50 mg/Kg qid for 11 days (1)	Targeted (1)– First-line (1)	Infection resolution	Association with minocycline. Implementation of a continuous infusion strategy with TDM
Falcone, 2021 [47]	Prospective cohort	Italy and Greece, November 2018–May 2019	52 (69)	39M, 13F	BSI (52)	*Klebsiella pneumoniae* (NAAG), *Escherichia coli* (NAAG), *Enterobacter species* (NAAG), and *Morganella morganii* (NAAG),	NDM and VIM	NAAG	CZA 2.5 g tid + ATM 2 g tid (1)	Targeted (52)– NAAG	Ten patients died out of 52	Original cohort made up of 102 patients, 69% males, median age 70, susceptibility rate of 88 % to COL. Median duration of antibiotic therapy 10 days (IQR 7–14)
Perrotta, 2021 [63]	Case report	Italy, December 2020	1 (57)	1M	BSI (1)	*Klebsiella pneumoniae* (1)	NDM	Susceptibility only to COL	CZA 1.25 g tid + ATM 1 g tid for 10 days (1)	Targeted (1)– First-line (1)	Clinical resolution and microbiological cure	Patient affected by TTP and COVID-19, with ICU admission due to severe interstitial pneumonia and with known rectal colonization by MBL-producing strain. At the end of therapy, the patient’s rectal swab tested negative, showing achievement of decolonization
Sieswerda, 2021 [64]	Case report	Netherlands, NR	1 (around 60 years)	1F	cUTI (1)	*Klebsiella pneumoniae* (1)	NDM-1	Pan-resistance	CZA 2.5 g tid + ATM 2 g tid for 14 days (1)	Targeted (1)– First-line (1)	After clinical resolution, the patient experienced a pyelonephritis by the same strain, recovered with an identical treatment	A case of BSI from urinary source in a kidney transplant recipient. Treatment commenced as empiric covering a previous bacteriuria by a carbapenemase-producing *K. pneumoniae* strain, then targeted according to blood culture results
Yasmin, 2021 [65]	Case report	United States, NR	1 (4)	1M	BSI (1)	*Enterobacter hormaechei subsp. hoffmannii* (1)	NDM-1 and KPC-4	Susceptibility only to COL	CZA 50 mg/Kg tid (as for ceftazidime component) + ATM 50 mg/Kg tid for 14 days (1)	Targeted (1) – Salvage (1)	Clinical and microbiological cure without relapse in 30-day follow-up	Pediatric patient with B cell acute lymphoblastic leukemia developing infection after stem cell infusion

AE: adverse event; AMK, amikacin; AST: antimicrobial susceptibility testing; ATM: aztreonam; BID: bis in die; BJI: bone and joint infection; BSI: bloodstream infection; CAZ: ceftazidime; CIED: cardiac implantable electronic device infection; CLABSI: central line-associated bloodstream infection; COL: colistin; COVID-19: coronavirus disease 2019; CRRT: continuous renal replacement therapy; CTX: cefotaximase; CZA: ceftazidime/avibactam; cUTI: complicated urinary tract infection (including pyelonephritis); F: female; HAP: hospital-acquired pneumonia; HSCT: hematopoietic stem cell transplantation; IMI: imipenem; IMP: imipenemase; IE: infective endocarditis; IQR: interquartile range; KPC: *Klebsiella pneumoniae* carbapenemase; MER: meropenem; M: male; MBL: metallo-β-lactamase; MIN: minocycline; NAAG: Not available due to aggregate data; NS: not specified; NDM: New Dehli metallo-beta-lactamase 1; NR: not reported; OD: once a day; OXA: Oxacillinase; POLB: Polymyxin B; QID: quarter in die; SOT: solid organ transplantation; TID: ter in die; TDM: therapeutic drug monitoring; TIG: tigecycline; TMP: trimethoprim/sulfamethoxazole; TTP: thrombocytopenic purpura; VAP: ventilator-associated pneumonia; VIM: Verona Integron-encoded metallo-β-lactamase; XDR: extensively drug-resistant. ^a^ Instead of the mean, the age of the single patient is provided with regard to case reports. ^b^ Type of therapy: empirical or targeted (evidence of in vitro synergy between ceftazidime/avibactam and aztreonam); line of therapy: first or salvage.

## Data Availability

All data pooled and analyzed for this systematic review and meta-analysis are included in the corresponding published articles, as reported in the main tables.

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
