# Peer review of "The Revival of Aztreonam in Combination with Avibactam against Metallo-β-Lactamase-Producing Gram-Negatives: A Systematic Review of In Vitro Studies and Clinical Cases"

_antibiotics, 2021, doi:10.3390/antibiotics10081012_

Round 1

Reviewer 1 Report

In this manuscript Mauri et al., conduct a literature review of in vitro and in vivo data about the potential use of Avibactam-Aztreonam combination for the treatment of bacteria carrying metallo-bactalactamases.

Several comments and clarification requests:

  1. Lines 14 and 15: “Aztreonam, an old beta-lactam antibiotic, is hydrolyzed by MBLs”. Mistake, it should be “NOT hydrolyzed”.
  2. in vitro should be in italics.
  3. In results section, when referring to enzyme production is really enzyme activity tested on those publications or is presence/absence of resistance genes? Nomenclature would be different (blaNDM vs NDM, for example). Lines 159 to 179 where enzyme production or activity are mentioned or is gene presence?
  4. Probably consider re-writing: line 232 “Overall, 94 patients were included in the present review, 97% adults and only three paediatric subjects”. 94 patients ARE included, instead of WERE. If 97% are adults this already means 3% are paediatrics, redundancy, no need for that? It could include percentage and actual number compared with total to be more specific if desired.
  5. Lines 349 to 353 seem a bit repetitive.
  6. Line 364: is “deserves” the correct verb?
  7. Lines 375-376 seem confusing.
  8. E. anopheles: Is relevant mentioning these bacteria?
  9. Line 409: This value overlapped with the OR. Present tense instead of past?

Author Response

In this manuscript Mauri et al., conduct a literature review of in vitro and in vivo data about the potential use of Avibactam-Aztreonam combination for the treatment of bacteria carrying metallo-beta-lactamases.

Several comments and clarification requests:

Lines 14 and 15: “Aztreonam, an old beta-lactam antibiotic, is hydrolyzed by MBLs”. Mistake, it should be “NOT hydrolyzed”.

Thank you. The sentence has been adjusted appropriately.

in vitro should be in italics.

 “In vitro” has been adjusted to italics all over the text.

In results section, when referring to enzyme production is really enzyme activity tested on those publications or is presence/absence of resistance genes? Nomenclature would be different (blaNDM vs NDM, for example). Lines 159 to 179 where enzyme production or activity are mentioned or is gene presence?

Thank you for your punctual suggestion. All publications regarding in vitro studies involved isolates positive for MBL genes characterized at the phenotypic level, thus evaluating resistance to carbapenems and detecting the resistance mechanism. It is noting that most of them were large surveillance studies involving isolates well characterized at genotypic and phenotypic level. In line with this context, we would prefer to maintain the enzyme nomenclature in order to refer to MBL determinants.

Probably consider re-writing: line 232 “Overall, 94 patients were included in the present review, 97% adults and only three paediatric subjects”. 94 patients ARE included, instead of WERE. If 97% are adults this already means 3% are paediatrics, redundancy, no need for that? It could include percentage and actual number compared with total to be more specific if desired.

The sentence has been rephrased, as required.

Lines 349 to 353 seem a bit repetitive.

The sentence has been rephrased, as required.

Line 364: is “deserves” the correct verb?

The term “deserves” has been changed with “may show”.

Lines 375-376 seem confusing.

The sentence has been rephrased, as required.

E. anopheles: Is relevant mentioning these bacteria?

E. anopheles is a rare opportunistic pathogen. We think that data regarding this species should be included in this review, for a complete overview of the studies and because all isolates belonging to this species were resistant to the combination ATM-AVI.

Line 409: This value overlapped with the OR. Present tense instead of past?

The present tense has been used, as suggested.

Reviewer 2 Report

To the authors

I believe this is a quite interesting review meta-analysis against the serious problem of possible alternatives to MDR infections caused by MBL-GN bacteria. 

Author Response

I believe this is a quite interesting review meta-analysis against the serious problem of possible alternatives to MDR infections caused by MBL-GN bacteria.

Thank you for your appreciation.

Reviewer 3 Report

The authors performed a systematic review of the literatures to evaluate the efficacy of aztreonam-avibactam combination against infection caused by MBL producing organisms. Since aztreonam-avibactam has been proposed as a major candidate for the treatment of MBL-producing Enterobacterales infection, several studies have been performed to evaluate its efficacy in vitro and in vivo. This review article is well-organized, and well-written, and the topic of this article could be interesting for the potential readers of this journal. However, I have a major concern in this review about in terms of vivo studies.

  1. A total of 18 studies were included in this meta-analysis, and 16 of them were case reports. In line 232, the author mentioned the characteristics of overall patients. However, majority of them were the study population of only three studies [Falcone 2020, Falcone 2021, and Shaw, 2018]. In addition, the inclusion of 16 case reports could result in a serious selection bias because only successive treatment experiences were published as case reports. Furthermore, the meta-analytic approach with only two studies does not have statistical power to reach to the real-world data. The review about in vivo studies should be limited to the introduction of cases and two recent original articles, and statistical analysis should be removed.
  2. The conclusion should be tone down that the in vitro activity of ATM-AVI were identified, but in vivo efficacy needs to be further evaluated. In addition, the need for further evaluation about in clinical efficacy of ATM-AVI could be suggested.

Author Response

The authors performed a systematic review of the literatures to evaluate the efficacy of aztreonam-avibactam combination against infection caused by MBL producing organisms. Since aztreonam-avibactam has been proposed as a major candidate for the treatment of MBL-producing Enterobacterales infection, several studies have been performed to evaluate its efficacy in vitro and in vivo. This review article is well-organized, and well-written, and the topic of this article could be interesting for the potential readers of this journal. However, I have a major concern in this review about in terms of in vivo studies.

Thank you for your appreciation.

A total of 18 studies were included in this meta-analysis, and 16 of them were case reports. In line 232, the author mentioned the characteristics of overall patients. However, majority of them were the study population of only three studies [Falcone 2020, Falcone 2021, and Shaw, 2018]. In addition, the inclusion of 16 case reports could result in a serious selection bias because only successive treatment experiences were published as case reports. Furthermore, the meta-analytic approach with only two studies does not have statistical power to reach to the real-world data. The review about in vivo studies should be limited to the introduction of cases and two recent original articles, and statistical analysis should be removed.

Thank you for your constructive comments. We are in agreement with the Reviewer about selection bias related to case reports and we have already discuss this limitation at the end of the Discussion section. Unfortunately, no prospective studies exist in the scientific literature, hence only observational in vivo studies have been included. To strength this concept we added: “Large prospective studies are urgently needed to better understand the in vivo efficacy of the ATM-AVI combination”.

We fully acknowledge the limitations of our meta-analytic approach, as carefully described in the discussion, that was a post-hoc analysis that has been added for the sake of clarity. The scenario of MBL infections in quite scattered worldwide. For sure, it affects some countries much more than others, and this explains why there are a few cohort studies and several case reports. Currently (and luckily, because the spread of the XDR strains would be catastrophic) there is no room for very large cohorts or properly conducted RCTs on the topic. This is why we humbly maintain that even a meta-analysis of only 2 studies, of course describing all its limitations, may be worthy of being shown, pursuant to the rule that to perform meta-analysis is possible as long as at least two studies are available (please see Boreinstein et al. Introduction to meta-analysis, first edition, Chapter 40: When Does it Make Sense to Perform a Me-ta-Analysis).

The conclusion should be tone down that the in vitro activity of ATM-AVI were identified, but in vivo efficacy needs to be further evaluated. In addition, the need for further evaluation about in clinical efficacy of ATM-AVI could be suggested.

We have added this sentence at the end of Discussion section: “At any rate further studies are needed to better define the clinical efficacy of CZA plus ATM, and only randomized clinical trials will provide high-quality evidence on the best therapeutic option for infections by MBL-producing strains”.

Round 2

Reviewer 3 Report

The authors answered all comments properly.

This article could be accepted for publication.